# Association of the platelet-to-high-density lipoprotein cholesterol ratio (PHR) with metabolic syndrome and metabolic overweight/obesity phenotypes: A study based on the Dryad database

Chuncheng Wu[1☯], Ziyi Hu[2☯], Ping Zhang[1¤*]

**1** The Second Department of Digestion, Affiliated Hospital of Jiangxi University of Chinese Medicine, Nanchang, Jiangxi, China, **2** Classical Department of Traditional Chinese Medicine, Affiliated Hospital of Jiangxi University of Chinese Medicine, Nanchang, Jiangxi, China

☯ These authors have contributed equally to this work and share first authorship.
¤Current Address: The Second Department of Digestion, Affiliated Hospital of Jiangxi University of Chinese Medicine, Nanchang, Jiangxi, China
* 1749743784@qq.com

## Abstract

### Background

Overweight/obesity and metabolic syndrome (MetS) are major global public health challenges. The platelet-to-high-density lipoprotein cholesterol ratio (PHR) has emerged as a potential biomarker reflecting both inflammatory status and lipid metabolism; however, its association with MetS and metabolic overweight/obesity phenotypes is not well understood. The present study aims to investigate the association of the PHR with MetS and metabolic overweight/obesity phenotypes.

### Methods

We derived data from the Dryad data repository. This retrospective study included 1,592 physical examination participants in Wuhan Union Hospital from 2020 to 2021. Based on BMI categories and metabolic status, we defined and distinguished four metabolic overweight/obesity phenotypes: metabolically healthy with normal weight (MHNW), metabolically unhealthy with normal weight (MUNW), metabolically healthy with overweight/obesity (MHO), and metabolically unhealthy with overweight/obesity (MUO). PHR was calculated as the ratio of platelets to HDL cholesterol. Logistic regression analysis was used to test the independent association between PHR with MetS and metabolic overweight/obesity phenotypes. In addition, we performed smooth curve fitting and subgroup analyses.

**Data availability statement:** Data are available in a public, open access repository. The datasets analyzed in this study are openly available in the Dryad Digital Repository (https://doi.org/10.5061/dryad.7d7wm3809). Citation: Yan, Fengqin; Nie, Guqiao; Zhou, Nianli et al. (2023). Association of fat-to-muscle ratio with non-alcoholic fatty liver disease: a single-centre retrospective study [Dataset]. Dryad. https://doi.org/10.5061/dryad.7d7wm3809

**Funding:** This work was supported by the 2022 Jiangxi Province Academic Experience Inheritance Project of Famous Traditional Chinese Medicine Experts.

**Competing interests:** The authors have declared that no competing interests exist.

**Abbreviations:** ALT, alanine aminotransferase; AST, aspartate aminotransferase; BMI, body mass index; CVD, cardiovascular diseases; DBP, diastolic blood pressure; FBG, fasting blood glucose; FMR, fat-to-muscle ratio; HDL-C, high-density lipoprotein-cholesterol; LDL-C, low-density lipoprotein-cholesterol; MAFLD, metabolic associated fatty liver disease; MetS, metabolic syndrome; MHNW, metabolically healthy with normal weight; MHO, metabolically healthy overweight/obesity; MUNW, metabolically unhealthy with normal weight; MUO, metabolically unhealthy overweight/obesity; PHR, platelet-to-high-density lipoprotein cholesterol ratio; PLT, platelet; SBP, systolic blood pressure; TC, total cholesterol; TG, triglyceride; UA, uric acid.

## Results

Logistic regression analysis, after adjusting for covariates, indicated that each 10-unit increase in PHR was associated with elevated risks of MetS (OR = 1.29, 95% CI: 1.24–1.35). There was a significant positive association between PHR and the occurrence of both MHO (OR = 1.15, 95% CI: 1.11–1.19) and MUO (OR = 1.16, 95% CI: 1.12–1.20), while the association with MUNW remained unclear (P = 0.73). In addition, we found a nonlinear relationship between PHR and the incidence of MetS, MHO, and MUO.

## Conclusion

PHR demonstrates strong associations with MetS, MHO, and MUO, indicating its potential utility as an early biomarker for metabolic dysfunction.

---

## 1. Introduction

Overweight/obesity represent major global health concerns, affecting more than 10% of adults worldwide [1]. In China, the prevalence of obesity and overweight among adults has reached 11.24% and 34.29%, respectively [2]. Overweight/obesity is a significant risk factor for the development and progression of metabolic syndrome (MetS) [3,4]. MetS assessment relies on multiple indicators, including blood pressure, blood glucose, blood lipids, and other parameters [5,6]. The metabolic status was assessed based on the presence or absence MetS [7]. Considering the association between BMI categories and metabolic status, individuals can be classified into one of four phenotypes: metabolically healthy normal weight (MHNW), metabolically unhealthy normal weight (MUNW), metabolically healthy overweight/obesity (MHO), and metabolically unhealthy overweight/obesity (MUO). The incidence of complications in MUO is significantly higher than in MHO. Studies of adipose tissue in these two phenotypes have revealed the metabolic heterogeneity of obesity and established a clear link between adipose tissue remodelling and metabolic disease development [8].

Overweight/obesity is a central component in MetS. Recent studies have demonstrated that even individuals with normal weight can exhibit MetS, frequently accompanied by abnormal body fat distribution [9]. Evidence indicates that MetS individuals face an increased risk of heart failure and left ventricular dysfunction compared to their metabolically healthy counterparts, even in non-obese populations [10]. These findings demonstrate that MetS are a significant health risk factor independent of body mass index (BMI) and are associated with increased all-cause mortality. Epidemiological studies in rural Chinese populations have documented similar associations between MetS and elevated mortality rates, independent of obesity [11]. These observations suggest that metabolically unhealthy status serves as a crucial mediator in disease development. Multiple studies have demonstrated that dynamic changes in metabolic health significantly influence disease outcomes. A prospective cohort study of 54,441 adults revealed that individuals with MHO at

baseline exhibited higher cardiovascular disease (CVD) incidence compared to MHNW individuals, with this association being more pronounced in younger age groups [12]. Furthermore, research in middle-aged and elderly Chinese populations has shown that deterioration in metabolic health increases CVD occurrence, while improvement to a healthy metabolic state reduces CVD susceptibility [13]. These findings highlight the importance of early identification and intervention strategies for metabolic health management.

The platelet-to-high-density lipoprotein cholesterol ratio (PHR) has emerged as a novel marker of inflammatory response and metabolic disorders. PHR, a composite index combining platelet count and HDL-C levels, reflects both systemic inflammatory status and thrombotic tendency [14]. Recent studies have demonstrated strong associations between PHR and various chronic conditions, including hypertension, heart failure, and diabetes, highlighting its potential as a predictor of cardiometabolic diseases [15–17]. Obesity-induced chronic low-grade inflammation represents a key risk factor in the pathogenesis and progression of metabolic disorders [4]. A retrospective study [18] demonstrated that PHR can be utilized as a valid marker in obese patients to reliably predict the presence of type 2 diabetes. Alice Marra et al. [19] showed that five inflammatory indices derived from complete blood cell counts, including PHR, have good validity in predicting MetS in severely obese adults, providing clinicians with useful tools to assess obesity-related metabolic risk. While earlier research has examined the association between PHR and metabolic diseases such as hypertension and diabetes, there remains a paucity of information regarding the relationship between PHR and MetS and its defined overweight/obesity phenotypes. Consequently, the present study aims to analyze the relationship between PHR and MetS, as well as metabolic obesity/overweight phenotypes, with the goal of providing deeper insights for clinical practice.

## 2. Methods

### 2.1. Study design and data source

This retrospective cross-sectional study utilized data from the Dryad digital repository (https://doi.org/10.5061/dryad.7d7wm3809). The study protocol was approved by the Ethics Committee of Wuhan Union Medical College Hospital. The study population comprised individuals who underwent comprehensive health examinations at Wuhan Union Medical College Hospital between January 2020 and November 2021. The detailed methodology and primary findings have been previously reported by Yan et al. [20].

### 2.2. PHR

The calculation method for PHR is the plasma platelet count ($10^9$/L) divided by the plasma high-density lipoprotein cholesterol level (mmol/L) [14].

### 2.3. Definitions of outcome

We defined the MetS (metabolically unhealthy status) as the appearance of ≥2 of these components according to the National Cholesterol Education Program Adult Treatment Panel-III (NCEP-ATP-III) criteria [21]: (1) Blood pressure: systolic ≥130 mmHg and/or diastolic ≥85 mmHg; (2) Fasting plasma glucose ≥5.6 mmol/L;(3) High-density lipoprotein cholesterol (HDL-C): Men: < 1.03 mmol/L, Women: < 1.29 mmol/L; (4) Triglycerides ≥1.7 mmol/L. The metabolic status was assessed based on the presence or absence MetS.

In the Chinese adult population, a BMI of less than 18.5 kg/m² indicates low weight status, a BMI between 18.5 kg/m² and less than 24 kg/m² indicates normal weight, a BMI between 24 kg/m² and less than 28 kg/m² indicates overweight, and a BMI of 28 kg/m² or more indicates obesity [22]. We decided to use the Chinese obesity standard to classify BMI in our study [23] because this standard aligns more closely with the characteristics of the Chinese population and health risk assessment.Patients were classified into BMI categories: normal weight (BMI < 24 kg/m²), and overweight/obesity (BMI ≥ 24 kg/m²).

The individuals were categorized into four groups according to their BMI and metabolic status [24]:

(1)  MHNW defined as BMI < 24 kg/m$^2$ and unhealthy metabolic status;

(2)  MHO defined as BMI ≥ 24 kg/m$^2$ and healthy metabolic status;

(3)  MUNW defined as BMI < 24 kg/m$^2$ and unhealthy metabolic status;

(4)  MUO defined as BMI ≥ 24 kg/m$^2$ and unhealthy metabolic status.

## 2.4. Data collection and variable definitions

The following variables were extracted from the database: age, gender, diastolic blood pressure (DBP) and systolic blood pressure (SBP), body mass index (BMI), fat-to-muscle ratio (FMR), tobacco use, alcohol use, platelet(PLT), alanine aminotransferase (ALT), aspartate aminotransferase (AST), fasting blood glucose (FBG), uric acid (UA), total cholesterol (TC), triglyceride (TG), low-density lipoprotein-cholesterol (LDL-C), high-density lipoprotein-cholesterol (HDL-C), hypertension, diabetes, and metabolic associated fatty liver disease (MAFLD). The age of the participants was categorised into the following groups: 40–49, 50–59, 60–69, and 70–79.

## 2.6. Statistical analysis

All statistical analyses were performed using R software (version 4.2.2) and EmpowerStats (version 4.2). Summaries of continuous baseline variables are presented as means and standard deviations unless skewed and then presented as medians and interquartile ranges (IQR). For normally distributed continuous variables, comparisons between groups were conducted using one-way analysis of variance. For non-normally distributed variables, Kruskal-Wallis tests were employed.

The independent association of PHR with MetS and metabolic overweight/obesity phenotypes was assessed using logistic regression analysis. To minimize potential bias and systematically evaluate the association between PHR and MetS (including metabolic overweight/obesity phenotypes), three sequential models were constructed: Model 1: unadjusted; Model 2: adjusted for sex, age, BMI, and FMR; Model 3: further adjusted for ALT, AST, UA, tobacco use, and alcohol use. Smooth curve fitting and threshold effect analysis were conducted using EmpowerStats (version 4.2) to investigate the relationship between PHR and MetS, as well as metabolic overweight and obesity phenotypes. Subgroup analyses using logistic regression models were conducted to assess the influence of population characteristics, with subjects stratified by gender, age, BMI, tobacco use, and alcohol use. Interaction effects between subgroups were evaluated, with interaction $P$-values calculated to determine the significance of characteristic-specific effects. Results are presented as forest plots, including odds ratios (OR) with 95% confidence intervals (CI), $P$-values, and interaction $P$-values for each subgroup. Statistical significance was set at a two-sided $P$-value < 0.05.

## 2.7. Ethical consideration

This study involves human participants and was approved by the Institutional Review Board of Tongji Medical College, Huazhong University of Science and Technology (S155). Informed consent was not required because all medical data were retrospectively reviewed and analysed anonymously.

## 3.  Results

### 3.1.  Baseline characteristics of the participants

This study included 1,592 participants (444 females, 1,148 males). Of these, 1,139 patients were diagnosed with MetS. The study population was further categorized into four groups: MHNW (n = 259), MHO (n = 268), MUNW(n = 194), and MUO (n = 871). Table 1 presents the baseline characteristics and biochemical parameters of participants stratified by PHR

**Table 1.  Clinical characteristics of the study participants by PHR tertiles.**

| Characters | Total | T1 group | T2 group | T3 group | P-value |
|---|---|---|---|---|---|
| PHR | (47.52-694.59) | <164.55 | 164.55-219.12 | >219.12 | |
| n | 1592 | 531 | 530 | 531 | |
| Age group, years, n(%) | | | | | <0.001 |
| 40-49 | 354 (22.24) | 91 (17.14) | 115 (21.70) | 148 (27.87) | |
| 50-59 | 709 (44.54) | 198 (37.29) | 259 (48.87) | 252 (47.46) | |
| 60-69 | 360 (22.61) | 163 (30.70) | 104 (19.62) | 93 (17.51) | |
| 70-79 | 169 (10.62) | 79 (14.88) | 52 (9.81) | 38 (7.16) | |
| Gender, n(%) | | | | | <0.001 |
| Female | 444 (27.89%) | 206 (38.79%) | 138 (26.04%) | 100 (18.83) | |
| Male | 1148 (72.11%) | 325 (61.21%) | 392 (73.96%) | 431 (81.17) | |
| BMI, kg/m$^2$ | 25.10 (23.30, 27.10) | 24.30 (22.60, 26.50) | 25.10 (23.60, 27.30) | 25.60 (23.80, 27.60) | <0.001 |
| FMR | 0.37 (0.31, 0.45) | 0.38 (0.31, 0.48) | 0.37 (0.32, 0.46) | 0.37 (0.32, 0.43) | 0.102 |
| Tobacco use, n (%) | | | | | <0.001 |
| No | 1054 (66.21) | 401 (75.52) | 343 (64.72) | 310 (58.38) | |
| Yes | 538 (33.79) | 130 (24.48) | 187 (35.28) | 221 (41.62) | |
| Alcohol use, n (%) | | | | | 0.004 |
| No | 1072 (67.34) | 386 (72.69) | 349 (65.85) | 337 (63.47) | |
| Yes | 520 (32.66) | 145 (27.31) | 181 (34.15) | 194 (36.53) | |
| Hypertension, n (%) | | | | | 0.003 |
| No | 649 (40.77) | 230 (43.31) | 234 (44.15) | 185 (34.84) | |
| Yes | 943 (59.23) | 301 (56.69) | 296 (55.85) | 346 (65.16) | |
| Diabetes, n (%) | | | | | <0.001 |
| No | 1094 (68.72) | 390 (73.45) | 372 (70.19) | 332 (62.52) | |
| Yes | 498 (31.28) | 141 (26.55) | 158 (29.81) | 199 (37.48) | |
| MAFLD, n (%) | | | | | <0.001 |
| No | 619 (38.88) | 271 (51.04) | 196 (36.98) | 152 (28.63) | |
| Yes | 973 (61.12) | 260 (48.96) | 334 (63.02) | 379 (71.37) | |
| MetS | | | | | <0.001 |
| No | 453 (28.45) | 263 (49.53) | 128 (24.15) | 62 (11.68) | |
| Yes | 1139 (71.55) | 268 (50.47) | 402 (75.85) | 469 (88.32) | |
| Metabolic overweight/obesity phenotypes | | | | | <0.001 |
| MHNW | 259 (16.27) | 158 (29.76) | 63 (11.89) | 38 (7.16) | |
| MHO | 268 (16.83) | 73 (13.75) | 95 (17.92) | 100 (18.83) | |
| MUNW | 194 (12.19) | 105 (19.77) | 65 (12.26) | 24 (4.52) | |
| MUO | 871 (54.71) | 195 (36.72) | 307 (57.92) | 369 (69.49) | |
| PLT, 10⁹/L | 207 (175, 241) | 174 (149, 202) | 206 (180, 233) | 241 (213, 280) | <0.001 |
| ALT, U/L | 21 (15, 30) | 19 (14, 26) | 22 (15, 30) | 24 (17, 36) | <0.001 |
| AST, U/L | 21 (17, 26) | 21 (17, 25) | 21 (17, 26) | 21 (17, 27) | 0.593 |
| FBG, mmol/L | 5.10 (4.70, 5.70) | 5.00 (4.60, 5.60) | 5.15 (4.70, 5.76) | 5.20 (4.80, 5.90) | <0.001 |
| UA, μmol/L | 363 (301, 426) | 335 (273, 396) | 369 (307, 435) | 378 (317, 445) | <0.001 |
| TC, mmol/L | 4.44 (3.73, 5.14) | 4.37 (3.62, 5.09) | 4.51 (3.80, 5.16) | 4.43 (3.75, 5.16) | 0.107 |
| TG, mmol/L | 1.42 (0.98, 2.18) | 1.03 (0.78, 1.48) | 1.49 (1.06, 2.15) | 1.89 (1.29, 2.90) | <0.001 |
| HDL-C, mmol/L | 1.07 (0.89, 1.28) | 1.33 (1.15, 1.54) | 1.07 (0.94, 1.20) | 0.88 (0.76, 1.01) | <0.001 |
| LDL-C, mmol/L | 2.65 (2.02, 3.22) | 2.54 (1.89, 3.09) | 2.73 (2.09, 3.30) | 2.67 (2.12, 3.26) | <0.001 |

*(Continued)*

**Table 1.** (Continued)

| Characters | Total | T1 group | T2 group | T3 group | *P*-value |
|---|---|---|---|---|---|
| SBP, mmHg | 130 (120, 140) | 128 (120, 140) | 130 (120, 140) | 130 (121, 141) | 0.032 |
| DBP, mmHg | 80 (74, 89) | 80 (73, 88) | 81 (74, 90) | 81 (76, 90) | <0.001 |

Continuous variables are presented as Median (IQR). Categorical variables are presented as n (%). Univariate logistic regression models were used for continuous and categorical variables. ALT = alanine aminotransferase, AST = aspartate aminotransferase, BMI = body mass index, DBP = diastolic blood pressure, FBG = fasting blood glucose, FMR = fat-to-muscle ratio, HDL-C = high-density lipoprotein-cholesterol, LDL-C = low-density lipoprotein-cholesterol, MetS = metabolic syndrome, MHNW = metabolically healthy with normal weight, MHO = metabolically healthy with overweight/obesity, MUNW = metabolically unhealthy with normal weight, MUO = metabolically unhealthy with overweight/obesity, MAFLD = metabolic associated fatty liver disease, PLT = platelet, SBP = systolic blood pressure, TC = total cholesterol, TG = triglyceride, UA = uric acid.

tertiles. The results indicated significant differences in clinical characteristics, including age, gender, BMI, and smoking and drinking habits, as PHR increased. Participants in the T3 group were older, had a lower proportion of women, and exhibited significantly higher BMI and platelet counts compared to those in the T1 group. Additionally, the incidence of MetS, MUO, and MAFLD was significantly higher in the T3 group.

### 3.2. Association between PHR and MetS

Table 2 demonstrates the associations between PHR and MetS across three statistical models. When analyzed as a continuous variable, each 10-unit increase in PHR was associated with elevated risks of MetS in all models (Model 1: OR = 1.15, 95% CI: 1.13–1.18; Model 2: OR = 1.16, 95% CI: 1.13–1.19; Model 3: OR = 1.29, 95% CI: 1.24–1.35). In the categorical analysis using PHR tertiles (T1: < 164.55, T2: 164.55–219.12, T3: > 219.12), compared to the lowest tertile, both middle and highest tertiles showed significantly increased risks. The T2 group demonstrated approximately three-fold increased risks across all models (Model 3: OR = 4.04, 95% CI: 2.85–5.73), while the highest tertile showed more pronounced associations (Model 1: OR = 7.42, 95% CI: 5.42–10.17; Model 2: OR = 8.18, 95% CI: 5.62–11.92; Model 3: OR = 16.10, 95% CI: 9.75–26.60). Trend tests were significant across all models (P < 0.0001), confirming a robust dose-dependent relationship between PHR and MetS, which persisted after comprehensive adjustment for potential confounders.

Fig 1 illustrates a nonlinear relationship between PHR and MetS after adjustment for potential confounders. Threshold effect analysis identified an inflection point at PHR = 273.53 (Table 3). Below this threshold, PHR demonstrated a strong positive association with the risk of MetS, characterized by a steep upward trend. Above 273.53, the relationship pla-teaued, showing a markedly attenuated rate of increase in MetS risk with further elevations in PHR.

**Table 2.** Associations between PHR and MetS.

| Variable | Model 1 | | Model 2 | | Model 3 | |
|---|---|---|---|---|---|---|
| | OR (95%CI) | *P*-value | OR (95%CI) | *P*-value | OR (95%CI) | *P*-value |
| PHR per 10 Units | 1.15 (1.13, 1.18) | <0.0001 | 1.16 (10.13, 1.19) | <0.0001 | 1.29 (1.24, 1.35) | <0.0001 |
| PHR tertile | | | | | | |
| T1(<164.55) | Ref. | | Ref. | | Ref. | |
| T2(164.55–219.12) | 3.08 (2.37, 4.00) | <0.0001 | 3.02 (2.21, 4.11) | <0.0001 | 4.04 (2.85, 5.73) | <0.0001 |
| T3(>219.12) | 7.42 (5.42, 10.17) | <0.0001 | 8.18 (5.62, 11.92) | <0.0001 | 16.10 (9.75, 26.60) | <0.0001 |
| *P* for trend | | <0.0001 | | <0.0001 | | <0.0001 |

Reference (Ref.); OR = Odds Ratio; CI = Confidence Interval; Model 1: unadjusted; Model 2: adjust for gender,age, BMI, and FMR. Model 3: adjust for: gender,age, BMI, FMR, ALT, AST, UA, tobacco use, and alcohol use.

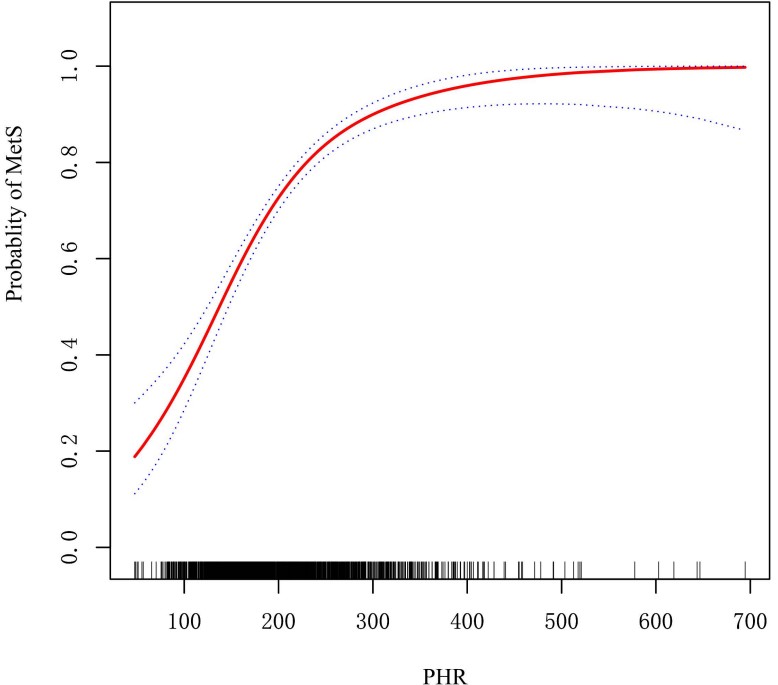

**Fig 1. Nonlinear relationship between PHR and incidence of MetS.** The relationship was modeled using restricted cubic splines with adjustment for gender,age, BMI, FMR, ALT, AST, UA, tobacco use, and alcohol use.

**Table 3. Threshold effect analysis of PHR on incidence of MetS.**

| Inflection point of PHR (U) | OR (95%CI)* | p value |
|---|---|---|
| <273.53 | 1.17(1.13, 1.21) | <0.0001 |
| ≥273.53 | 1.05 (0.97, 1.14) | 0.25 |
| Logarithmic likelihood ratio test *P* value | | 0.047 |

*PHR per 10 units; adjust for: gender,age, FMR, ALT, AST, UA, tobacco use, and alcohol use.

### 3.3. Association between PHR and metabolic overweight/obese phenotypes

Fig 2 demonstrate the distribution of PHR across different metabolic overweight/obese phenotypes. The MUO group exhibited the highest PHR level (221.05), while the MHNW group showed the lowest (160.55). The results of the logistic regression analysis are presented in Table 4. The findings indicate that for every 10-unit increase in PHR, the likelihood of MHO occurrence significantly increases (Model 1: OR = 1.16, 95%CI = 1.12–1.19; Model 2: OR = 1.17, 95%CI = 1.13–1.22; Model 3: OR = 1.15, 95%CI = 1.11–1.19). In the quantile analysis of PHR, the OR for the T2 and T3 groups were 3.26 (95% CI: 2.14–4.98) and 5.70 (95% CI: 3.58–9.07), respectively, indicating a significant increase in risk. Conversely, the association between PHR and the occurrence of MUNW was not significant, with P-values for all models exceeding 0.05. In contrast, the association between PHR and MUO was significant. For each 10-unit increase in PHR, the risk of MUO occurrence significantly increased, with an OR of 1.17 (95% CI: 1.14–1.20). The OR for the T2 and T3 groups were 3.95 (95% CI: 2.80–5.56) and 7.87 (95% CI: 5.30–11.67), respectively. These results indicate a significant positive association between PHR and the occurrence of both MHO and MUO, while the association with MUNW remains unclear.

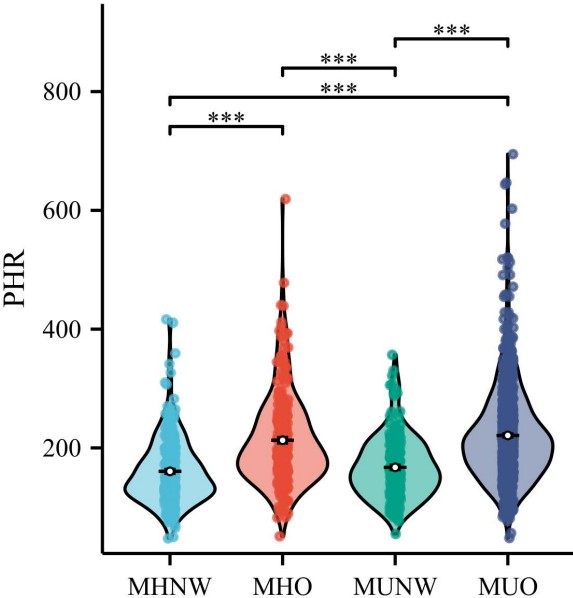

**Fig 2. Distribution of platelet-to-HDL ratio (PHR) across metabolically healthy obesity phenotypes.** Violin plots display the distribution patterns, medians, and quartiles for MHNO, MHO, MUNO, and MUO groups. ***$P < 0.001$ by *Kruskal-Wallis* test.

**Table 4. Multivariate logistic analysis associations between PHR and metabolic overweight/obese phenotypes.**

| Variable | Model1 | | Model 2 | | Model 3 | |
|---|---|---|---|---|---|---|
| | OR (95%CI) | *P*-value | OR (95%CI) | *P*-value | OR (95%CI) | *P*-value |
| **Incidence of MHO** | | | | | | |
| PHR per 10 Units | 1.16(1.12, 1.19) | <0.0001 | 1.17 (1.13, 1.22) | <0.0001 | 1.15(1.11, 1.19) | <0.0001 |
| PHR tertile | | | | | | |
| T1(<164.55) | Ref. | | Ref. | | Ref. | |
| T2(164.55–219.12) | 3.26 (2.14, 4.98) | <0.0001 | 3.59 (2.24, 5.75) | <0.0001 | 3.28 (2.01, 5.34) | <0.0001 |
| T3(>219.12) | 5.70 (3.58, 9.07) | <0.0001 | 7.80 (4.59, 13.27) | <0.0001 | 6.25 (3.62, 10.78) | <0.0001 |
| **Incidence of MUNW** | | | | | | |
| PHR per 10 Units | 1.03(0.99, 1.06) | 0.17 | 1.02 (0.98, 1.07) | 0.315 | 1.01 (0.96, 1.05) | 0.73 |
| PHR tertile | | | | | | |
| T1(<164.55) | Ref. | | Ref. | | Ref. | |
| T2(164.55–219.12) | 1.55 (1.01, 2.38) | 0.04 | 1.32 (0.79, 2.19) | 0.29 | 1.24 (0.74, 2.08) | 0.42 |
| T3(>219.12) | 0.95 (0.54, 1.68) | 0.86 | 1.07 (0.56, 2.07) | 0.83 | 0.93 (0.48, 1.81) | 0.83 |
| **Incidence of MUO** | | | | | | |
| PHR per 10 Units | 1.17(1.14, 1.20) | <0.0001 | 1.18(1.14, 1.22) | <0.0001 | 1.16(1.12, 1.20) | <0.0001 |
| PHR tertile | | | | | | |
| T1(<164.55) | Ref. | | Ref. | | Ref. | |
| T2(164.55–219.12) | 3.95 (2.80, 5.56) | <0.0001 | 3.45 (2.16, 5.49) | <0.0001 | 3.10(1.92, 5.02) | <0.0001 |
| T3(>219.12) | 7.87 (5.30, 11.67) | <0.0001 | 9.08 (5.29, 15.59) | <0.0001 | 7.15 (4.14, 12.36) | <0.0001 |

Reference (Ref.);OR = Odds Ratio; CI = Confidence Interval; Model 1: unadjusted; Model 2: adjust for gender,age, and FMR. Model 3: adjust for: gender,age, FMR, ALT, AST, UA, tobacco use, and alcohol use.

Using generalized additive models and restricted cubic spline analysis, we further explored the correlation between PHR and the risk of MHO or MUNO. Fig 3 illustrates a nonlinear relationship between PHR and incidence of MHO (Fig 3A) and MUO (Fig 3B) after adjustment for potential confounders. We identified inflection points at 166.67 for MHO and 236.36 for MUO, respectively. When PHR was below 166.67, the risk of MHO increased by 28% (P<0.0001). When PHR was ≥ 166.67, the increase in MHO risk significantly slowed (P<0.0001). When PHR was below 236.36, the risk of MUO increased by 21% (P<0.0001). When PHR was ≥ 236.36, it was no longer significantly associated with MUO (P=0.159). Refer to Table 5 for detailed information.

### 3.4. Subgroup analyses

Subgroup analyses revealed consistent positive associations between PHR and MetS across diverse population characteristics, including demographic factors (gender, age), anthropometric measures (BMI), lifestyle behaviors (smoking,

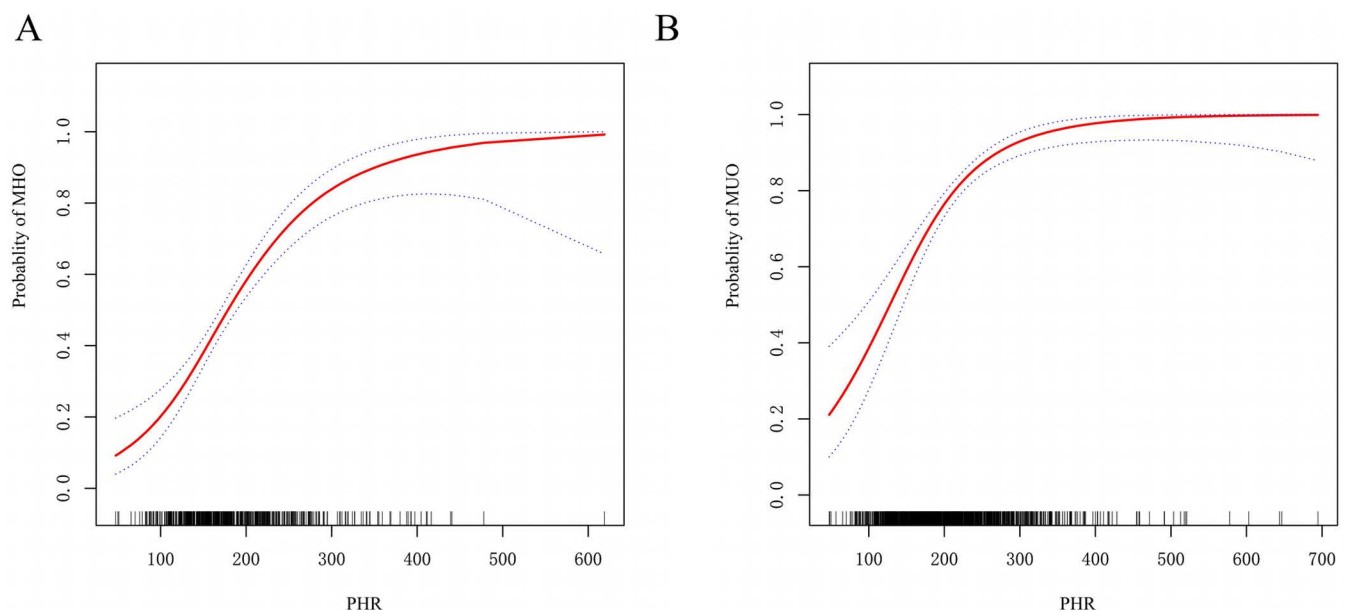

**Fig 3. Nonlinear relationship between PHR and incidence of MHO (A) and MUO (B).** The relationship was modeled using restricted cubic splines with adjustment for gender,age, BMI, FMR, ALT, AST, UA, tobacco use, and alcohol use.

**Table 5. Threshold effect analysis of PHR on incidence of MHO and MUNO.**

| Inflection point of PHR (U) | OR (95%CI)* | p value |
|---|---|---|
| **MHO** | | |
| <166.67 | 1.28(1.14, 1.44) | <0.0001 |
| ≥166.67 | 1.12 (1.06, 1.18) | <0.0001 |
| Logarithmic likelihood ratio test *P* value | | 0.056 |
| **MUO** | | |
| <236.36 | 1.21 (1.13, 1.28) | <0.0001 |
| ≥236.36 | 1.07 (0.97, 1.18) | 0.159 |
| Logarithmic likelihood ratio test *P* value | | 0.094 |

*PHR per 10 units; adjust for: gender,age, FMR, ALT, AST, UA, tobacco use, and alcohol use.

| Variables | n (%) | OR (95%CI) | | P | P for interaction |
|---|---|---|---|---|---|
| All patients | 1592 (100.00) | 1.15 (1.12 ~ 1.18) | | <0.001 | |
| Gender | | | | | 0.600 |
| Female | 444 (27.89) | 1.15 (1.10 ~ 1.21) | | <0.001 | |
| Male | 1148 (72.11) | 1.15 (1.11 ~ 1.19) | | <0.001 | |
| Age | | | | | 0.085 |
| 40-49 | 354 (22.24) | 1.21 (1.13 ~ 1.30) | | <0.001 | |
| 50-59 | 709 (44.54) | 1.14 (1.10 ~ 1.19) | | <0.001 | |
| 60-69 | 360 (22.61) | 1.11 (1.05 ~ 1.17) | | <0.001 | |
| 70-79 | 169 (10.62) | 1.22 (1.11 ~ 1.33) | | <0.001 | |
| Tobacco use | | | | | 0.402 |
| No | 1054 (66.21) | 1.16 (1.12 ~ 1.20) | | <0.001 | |
| Yes | 538 (33.79) | 1.13 (1.08 ~ 1.19) | | <0.001 | |
| Alcohol use | | | | | 0.389 |
| No | 1072 (67.34) | 1.15 (1.11 ~ 1.18) | | <0.001 | |
| Yes | 520 (32.66) | 1.16 (1.11 ~ 1.23) | | <0.001 | |
| BMI Group | | | | | 0.806 |
| <24 | 527 (33.10) | 1.16 (1.11 ~ 1.21) | | <0.001 | |
| ≥24 | 1065 (66.90) | 1.15 (1.10 ~ 1.19) | | <0.001 | |

1.0   1.1   1.2   1.3

**Fig 4. Forest plot of stratified analyses for the association between PHR and MetS.** OR and 95%CI are shown for each 10-unit increase in PHR, stratified by demographic and clinical characteristics. Analyses were adjusted for potential confounders except for the stratification variable. The vertical dashed line represents OR = 1.0. *P*-interaction values indicate the significance of effect modification between subgroups.

alcohol consumption). The strength of these associations remained relatively stable across all stratified analyses. Notably, interaction tests showed no significant effect modification by any of these variables (all *P*-interaction > 0.05), suggesting the robustness of the PHR-MetS relationship across different population subgrups (See Fig 4).

## 4. Discussion

This retrospective study of 1,592 participants investigated the association between PHR and MetS, with particular focus on metabolically overweight/obesity phenotypes. Our findings revealed several key insights:(1) A significant positive association was identified between PHR and MetS, with analyses demonstrating a notable increase in the risk of MetS for each 10-unit increase in PHR across all statistical models. In the quantile analysis, both the middle quantile (T2) and the highest quantile (T3) exhibited a significant increase in risk compared to the lowest quantile (T1). Threshold effect analysis revealed an inflection point at PHR = 273.53. (2) The distribution of PHR varied among different metabolic overweight/obesity phenotypes, with the MUO group exhibiting the highest PHR level (221.05) and the MHNW group showing the lowest (160.55). (3) The logistic regression analysis indicated a positive association between PHR and both MHO and MUO, while the relationship between PHR and MUNW was less defined. Utilizing generalized additive models and restricted cubic spline analyses, we further explored the association between PHR and the risk of MHO and MUNW. The findings indicated a non-linear relationship between PHR and both MHO and MUO, with identified inflection points at 166.67 and 236.36, respectively. Collectively, these findings suggest that PHR may serve as a potential early warning indicator for metabolic abnormalities.

PHR has emerged as a promising inflammatory marker, attracting increasing attention due to its cost-effectiveness, accessibility, and stability compared to absolute blood cell counts. The biological rationale for PHR stems from its components: platelets serve as crucial inflammatory mediators and can initiate the intrinsic coagulation cascade, while HDL-C demonstrates antithrombotic properties through modulation of platelet function and coagulation pathways [25–27]. In the pioneering study by Jialal et al., which compared 58 metabolic syndrome patients with 44 healthy controls, PHR showed

significant elevation in patients and superior diagnostic value over traditional indicators for cardiovascular risk prediction [28]. Subsequent research has further validated PHR's clinical utility: Guo et al.'s analysis of NHANES data (2005–2018) revealed significant associations with diabetes and prediabetes [17], Chen et al. demonstrated its relationship with hypertension [15], and Wang et al. identified its potential as a biomarker for heart failure [16]. These findings collectively support PHR's value in predicting metabolic-related diseases. Our study's distinctive contribution lies in applying the metabolic obesity phenotypes framework proposed by Tsatsoulis et al. [7]. We observed significantly elevated PHR levels in both MUO and MHO groups compared to MHNW, suggesting that both metabolic dysfunction and obesity may contribute to increased PHR levels through enhanced inflammatory responses. While the MUNW group showed higher PHR levels than MHNW, this difference did not reach statistical significance, possibly due to limited sample size or individual variability. The absence of significant differences between MHNW and MUNW groups may reflect their shared non-obese status, suggesting that metabolic abnormalities alone might have limited impact on PHR levels or their effects might be confounded by other factors.

Obesity is a central component in metabolic disorders. With advancing research, a considerable proportion of obese individuals have been found to be metabolically healthy. Karelis et al. [29] pioneered the concept of MHO, describing individuals who are obese but metabolically normal. However, a specific definition of metabolic health remains inconsistent [30,31]. In this study, NCEP-ATPIII criteria were used to define metabolic health status. Numerous studies [12,13] suggest that MHO may be a transient stage in obesity progression. Further studies are needed to clarify the underlying mechanisms. The nonlinear relationship between PHR and metabolically unhealthy status may reflect complex pathophysiological mechanisms at different stages, including platelet activation, lipid metabolism disorders, and chronic inflammation. Mir et al. [4] revealed characteristic changes in inflammatory proteins in patients with obesity and MetS, suggesting that platelet activation and abnormal lipoprotein metabolism may underlie the association between PHR and MUO. The relationship between human obesity and inflammation was first revealed in 1995 [32], who found that TNF-α mRNA expression in obese adipose tissue was 2.5 times higher than in normal-weight individuals. Activated platelets release chemerin, an adipokine involved in inflammation, obesity, insulin resistance, and MetS [33,34]. HDL-C, as a multifunctional lipoprotein, has antioxidant and anti-inflammatory properties. Reduced HDL-C may attenuate its functional and antiatherogenic effects through decreased cholesterol efflux and hepatic uptake, as well as diminished antioxidant, anti-inflammatory, and anti-thrombotic effects [35,36]. The combination of increased platelets and decreased HDL-C may compromise these protective effects, promoting metabolic abnormalities.

This study importantly establishes the association between PHR and metabolic phenotypes and identifies a clinically meaningful threshold. These findings have significant clinical translational value: PHR can serve as a simple, economical biomarker for early metabolic risk screening; the established threshold provides a reference point for clinical intervention; and this metric may help identify metabolically unhealthy normal-weight individuals. Mechanistically, PHR reflects both inflammatory status and lipid metabolism dysfunction, offering a novel perspective for early detection of metabolic abnormalities.

## 5. Limitations and future directions

Several limitations of our study warrant consideration. First, the single-center retrospective design may introduce selection bias, potentially limiting the generalizability of our findings. Second, the cross-sectional nature of our study precludes establishment of causal relationships between PHR and metabolic outcomes. Future research directions should include: (1) Multicenter prospective studies to validate PHR's prognostic value and establish causality; (2) Mechanistic investigations to elucidate the biological pathways linking PHR to metabolic dysfunction; (3) Intervention studies to evaluate PHR's utility in monitoring treatment responses; (4) Development of personalized prevention strategies based on PHR thresholds. These efforts would strengthen the scientific foundation for PHR's clinical application in metabolic risk assessment and intervention.

## 6. Conclusion

PHR demonstrates strong associations with MetS, MHO, and MUO, indicating its potential utility as an early biomarker for metabolic dysfunction.

## Acknowledgments

The authors would like to thank the investigators of Tongji Medical College of Huazhong University of Science and Technology for sharing their data.

## Author contributions

**Conceptualization:** Ping Zhang.

**Data curation:** Ziyi Hu.

**Methodology:** Ziyi Hu.

**Software:** Ziyi Hu.

**Writing – original draft:** Chuncheng Wu, Ziyi Hu, Ping Zhang.

**Writing – review & editing:** Chuncheng Wu, Ping Zhang.

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
