## [Decision Letter · Decision Letter 0]

11 Feb 2025

PONE-D-25-00775Platelet-to-high-density lipoprotein cholesterol ratio and risk of metabolically unhealthy phenotype: A study based on the Dryad databasePLOS ONE

Dear Dr. Zhang,

Thank you for submitting your manuscript to PLOS ONE. After careful consideration, we feel that it has merit but does not fully meet PLOS ONE’s publication criteria as it currently stands. Therefore, we invite you to submit a revised version of the manuscript that addresses the points raised during the review process.

We look forward to receiving your revised manuscript.

Kind regards,

Marwan Al-Nimer

Academic Editor

PLOS ONE

4. In the online submission form, you indicated that your data will be submitted to a repository upon acceptance.  We strongly recommend all authors deposit their data before acceptance, as the process can be lengthy and hold up publication timelines. Please note that, though access restrictions are acceptable now, your entire minimal  dataset will need to be made freely accessible if your manuscript is accepted for publication. This policy applies to all data except where public deposition would breach compliance with the protocol approved by your research ethics board. If you are unable to adhere to our open data policy, please kindly revise your statement to explain your reasoning and we will seek the editor's input on an exemption.

Additional Editor Comments:

Dear

This retrospective study assess the PHR in normal built, overweight, and obese people with / without metabolic disturbance. The following comments required explanation:

1: Typing errors e.g. reference

2: The ATP III is the criteria of metabolic syndrome. Why this scientific term is not used?

3: NAFLD is an old term. It is substituted with metabolic dysfunction associated with liver disease

4: The diagnostic criteria of body mass index is not according to the WHO criteria. Please add a reference of using the criteria of BMI that appeared in the article

Regards

Reviewers' comments:

Reviewer's Responses to Questions

**Comments to the Author**

1. Is the manuscript technically sound, and do the data support the conclusions?

Reviewer #1: Partly

Reviewer #2: Partly

2. Has the statistical analysis been performed appropriately and rigorously? 

Reviewer #1: No

Reviewer #2: Yes

3. Have the authors made all data underlying the findings in their manuscript fully available?

Reviewer #1: Yes

Reviewer #2: No

4. Is the manuscript presented in an intelligible fashion and written in standard English?

Reviewer #1: Yes

Reviewer #2: Yes

5. Review Comments to the Author

Reviewer #1: I have read the article titled: “Platelet-to-high-density lipoprotein cholesterol ratio and risk of metabolically unhealthy phenotype: A study based on the Dryad database “ which is about a retrospective cross-sectional study that investigated the association of the platelet-to high-density lipoprotein cholesterol ratio (PHR) with metabolic health status and phenotypes of metabolic health status according to BMI. Some concerns should be addressed by the authors to improve consistency of the study

Main concerns:

1. The objective of the study is not stated clearly in the introduction. The authors should ameliorate this statement and show what is exactly known about PHR, obesity and metabolically unhealthy status.

2. The scientific terminology is inappropriately reformulated thorough the text leading to confusions. For example, instead of saying “Classification of metabolic overweight /obesity phenotypes” use simple terms like: classification according to metabolic status and BMI.

3. The research methodology is misleading because of the choice of potential confounding factors for adjustments. Some confounders used by the authors cannot be consider as such because they were already used to classify the participants as MU or MH (Adult Treatment Panel III (ATP III) criteria (Blood pressure, diastolic and systolic, Fasting plasma glucose; High-density lipoprotein cholesterol (HDL-C), and Triglycerides) or to classify the participant as Obese and non-obese (BMI). So, they cannot be used as confounders. Hence,

o In sections 3.2 and 3.3, as well as in the corresponding Tables 2 and 3, only two models should be kept: the unadjusted model and a model adjusted for these factors: gender, age, BMI, FMR. ALT, AST, UA, tobacco use, alcohol use and NAFLD.

o The same comment can be done for Table 4 regarding the adjustment.

o In subgroup analyses, do not include comorbidity conditions (hypertension, diabetes and NAFLD) because they are already associated to MU status.

4. The comparison of PHR across the 4 phenotypes (Figure 1) should be done by ANOVA test because the Kruskal-Wallis test is a non-parametric test and the sample size is considerable.

Minor concerns:

• Table 1: please mention in the title “by PHR tertiles”

• Put the section 3.3 (Smooth curve fitting and threshold effect analysis) after the section 3. 2

• Use the same abbreviations: sometimes we can read MHNW and sometimes MHNO.

• In section 3.3 and table 4 title, mention MU “probability”

• In section 3.4, figure 3 is not mentioned.

Reviewer #2: 90 This retrospective cross-sectional study utilized data from the Dryad digital

91 repository (doi : 10.5061/dryad.7d7wm3809). This design poses a problem: is it a cross-sectional analytical study? Or a retrospective case-control study?

A cross-sectional study is, par excellence, a prevalence study, taking into account that the study population included people who underwent full health examinations at Wuhan Union Medical College Hospital between January 2020 and November 2021. Clearly, the survey was not carried out in the community, but the data was hospital-based. What did you do to minimise this bias in relation to a cross-sectional design?

You just mentioned the study population. Did you draw a sample from this population, and if so, how did you do it?

134 EmpowerStats (version 4.2). Continuous variables are presented as mean ± standard

135 deviation, and categorical variables as numbers (percentages). And what did you do for the variables whose distribution was not normal?

139 The independent association between PHR and metabolic health status (including

140 metabolically healthy overweight/obesity phenotypes) was assessed using logistic

141 regression analysis. Logistic regression involves testing a regression model in which the dependent variable is dichotomous (coded 0-1). In this study, can you clarify what the dependent variable is and how it works?

6. PLOS authors have the option to publish the peer review history of their article (what does this mean? ). If published, this will include your full peer review and any attached files.

**Do you want your identity to be public for this peer review?** For information about this choice, including consent withdrawal, please see our Privacy Policy .

Reviewer #1: No

Reviewer #2: No

---

## [Author Response · Author response to Decision Letter 1]

9 Mar 2025

Responses to the Editor’s comments

Comment 1: Typing errors e.g. reference

Response: Thank you for bringing the typing errors to my attention. I have thoroughly reviewed the manuscript and corrected all identified errors. Your feedback is greatly appreciated, and I strive to ensure the accuracy and clarity of the manuscript.

Comment 2: The ATP III is the criteria of metabolic syndrome. Why this scientific term is not used?

Response: Thank you for your review of our paper and your valuable comments.The diagnostic criteria for metabolic syndrome define metabolic abnormalities through a combination of indicators rather than as a single disease. This classification aligns better with its pathological complexity, dynamic evolution, and clinical intervention needs. In the context of obesity, the concept of metabolic health has garnered significant attention in the scientific community. The concept of metabolic health is not exclusive to obesity; it is increasingly used to assess the risk of cardiovascular disease and diabetes-related complications[1,2].

The predominant approach in research defines metabolic health as the absence of metabolic syndrome. For example, a 2019 systematic review identified 35 cohort studies on cardiovascular disease risk, 21 of which used definitions of metabolic health associated with metabolic syndrome[3]. Eleven definitions were derived from the National Cholesterol Education Program Adult Treatment Panel III (NCEP ATPIII)[4], five from the International Diabetes Federation (IDF)[5], and an additional five were based on the uniform international definition of metabolic syndrome MetS) [6].

The NCEP ATP III criteria were used to define metabolic status. According to these criteria, unhealthy metabolic status MetS) was defined as the presence of two or more relevant components.Based on BMI categories and metabolic status�we defined and distinguished four metabolic overweight/obesity phenotypes: metabolically healthy with normal weight (MHNW), metabolically unhealthy with normal weight (MUNW), metabolically healthy with overweight/obesity (MHO), and metabolically unhealthy with overweight/obesity (MUO). This classification approach is becoming increasingly common in studies aimed at highlighting various metabolic phenotypes of overweight/obesity and their impact on health outcomes.

For example, in the study by Hwi Seung Kim et al. [7], the NCEP ATP III criteria were used to define metabolic status and participants were divided into metabolically healthy non-obese group, metabolically unhealthy non-obese group, metabolically healthy obese group, and metabolically unhealthy obese group. In the study by Zinuo Yuan et al.[8], which also considered BMI categories and metabolic status, four metabolic overweight/obesity phenotypes were distinguished: MHNW, MUNW, MHO and MUO.

To enhance the communication of the core content and findings of our study, we considered implementing the following major changes:

(1)Revised Title: Association of the platelet-to-high-density lipoprotein cholesterol ratio with metabolic syndrome and metabolic overweight/obesity phenotypes: A study based on the Dryad database.

(2)Revised Introduction: Overweight/obesity is a significant risk factor for the development and progression of metabolic syndrome (MetS) [3,4]. MetS assessment relies on multiple indicators, including blood pressure, blood glucose, blood lipids, and other parameters [5,6].The metabolic status was assessed based on the presence or absence MetS [7]. Considering the association between BMI categories and metabolic status, individuals can be classified into one of four phenotypic groups: metabolically healthy normal weight (MHNW), metabolically unhealthy normal weight (MUNW), metabolically healthy overweight/obese (MHO), and metabolically unhealthy overweight/obese (MUO).

(3)Methods and Results :The methods and results section has been reanalyzed and rewritten. Please refer to the manuscript for further details.

Additionally, we identified some terminology confusion in the manuscript and made revisions.

We look forward to further feedback to refine this work.

References

1.Stefan N, Häring HU, Hu FB, Schulze MB. Metabolically healthy obesity: epidemiology, mechanisms, and clinical implications. Lancet Diabetes Endocrinol. 2013 Oct;1(2):152-62. doi: 10.1016/S2213-8587(13)70062-7.

2.Tsatsoulis A, Paschou SA. Metabolically Healthy Obesity: Criteria, Epidemiology, Controversies, and Consequences. Curr Obes Rep. 2020 Jun;9(2):109-120. doi: 10.1007/s13679-020-00375-0.

3.Yeh TL, Chen HH, Tsai SY, Lin CY, Liu SJ, Chien KL. The Relationship between Metabolically Healthy Obesity and the Risk of Cardiovascular Disease: A Systematic Review and Meta-Analysis. J Clin Med. 2019 Aug 15;8(8):1228.

doi: 10.3390/jcm8081228.

4.Detection EPo, Adults ToHBCi. Executive Summary of The Third Report of The National Cholesterol Education Program (NCEP) Expert Panel on Detection, Evaluation, And Treatment of High Blood Cholesterol In Adults (Adult Treatment Panel III). JAMA. 2001 May 16;285(19):2486-97. doi: 10.1001/jama.285.19.2486.

Alberti KGMM, Zimmet P, Shaw J. Metabolic syndrome--a new world-wide definition. A Consensus Statement from the International Diabetes Federation. Diabet Med. 2006 May;23(5):469-80. doi: 10.1111/j.1464-5491.2006.01858.x.

5.Alberti KGMM, Eckel RH, Grundy SM, Zimmet PZ, Cleeman JI, Donato KA, et al. Harmonizing the metabolic syndrome: a joint interim statement of the International Diabetes Federation Task Force on Epidemiology and Prevention; National Heart, Lung, and Blood Institute; American Heart Association; World Heart Federation; International Atherosclerosis Society; and International Association for the Study of Obesity. Circulation. 2009 Oct 5;120(16):1640-5.

doi: 10.1161/CIRCULATIONAHA.109.192644.

6.Kim HS, Lee J, Cho YK, Park JY, Lee WJ, Kim YJ, et al. Differential Effect of Metabolic Health and Obesity on Incident Heart Failure: A Nationwide Population-Based Cohort Study. Front Endocrinol (Lausanne). 2021 Feb 25;12:625083. doi: 10.3389/fendo.2021.625083.

7.Yuan Z, Cheng Y, Han J, Wang D, Dong H, Shi Y, et al. Association between metabolic overweight/obesity phenotypes and readmission risk in patients with lung cancer: A retrospective cohort study. EClinicalMedicine. 2022 Jul 22;51:101577. doi: 10.1016/j.eclinm.2022.101577.

Comment 3: NAFLD is an old term. It is substituted with metabolic dysfunction associated with liver disease

Response: We sincerely appreciate your review of our paper and your valuable comments. In 2020, the International Fatty Liver Expert Group recommended renaming non-alcoholic fatty liver disease (NAFLD) to metabolic dysfunction-associated fatty liver disease (MAFLD) to better reflect its relationship with metabolic health [1] . This amendment aims to highlight the pivotal role of metabolic dysfunction in the etiology of fatty liver disease. It is important to note that the term NAFLD has been replaced by MAFLD. We have updated the relevant terminology in the manuscript.

References

1. Eslam M, Newsome PN, Sarin SK, Anstee QM, Targher G, Romero-Gomez M, et al. A new definition for metabolic dysfunction-associated fatty liver disease: An international expert consensus statement. J Hepatol. 2020 Jul;73(1):202-209. doi: 10.1016/j.jhep.2020.03.039.

Comment 4: The diagnostic criteria of body mass index is not according to the WHO criteria. Please add a reference of using the criteria of BMI that appeared in the article.

Response: The World Health Organization's classical cutoff values for normal weight (18.5-24.9 kg/m²), overweight (25.0-29.9 kg/m²), and obesity (≥30.0 kg/m²) were used, while lower cutoff values were applied for the Asian population. A study by the Pan XF et al. published in 2021 in The Lancet Diabetes & Endocrinology noted that in the Chinese adult population, a BMI of less than 18.5 kg/m² indicates low weight, a BMI between 18.5 kg/m² and less than 24 kg/m² indicates normal weight, a BMI between 24 kg/m² and less than 28 kg/m² indicates overweight, and a BMI of 28 kg/m² or more indicates obesity [1]. We decided to use the Chinese obesity standard to classify BMI in our study [2] because this standard aligns more closely with the characteristics of the Chinese population and health risk assessment.

Modified:

In the Chinese adult population, a BMI of less than 18.5 kg/m² indicates low weight status, a BMI between 18.5 kg/m² and less than 24 kg/m² indicates normal weight, a BMI between 24 kg/m² and less than 28 kg/m² indicates overweight, and a BMI of 28 kg/m² or more indicates obesity [20]. We decided to use the Chinese obesity standard to classify BMI in our study [21] because this standard aligns more closely with the characteristics of the Chinese population and health risk assessment.Patients were classified into BMI categories: normal weight (BMI <24 kg/m2), and overweight/obesity (BMI ≥24 kg/m2).

References

1. Pan XF, Wang L, and Pan A. Epidemiology and determinants of obesity in China. Lancet Diabetes Endocrinol. (2021) 9:373-392.

doi: 10.1016/S2213-8587(21)00045-0.

2.National Clinical Practice Guideline on Obesity Management Editorial Committee .National Clinical Practice Guideline on Obesity Management (2024 Edition).Chinese Circulation Journal. (2025)40: 6.

doi: 10.3969/j.issn.1000-3614.2025.01.002

Responses to the Reviewer #1’s comments

Comment 1: The objective of the study is not stated clearly in the introduction. The authors should ameliorate this statement and show what is exactly known about PHR, obesity and metabolically unhealthy status.

Response: Thank you for your constructive comments! We have made the following changes:

The platelet-to-high-density lipoprotein cholesterol ratio (PHR) has emerged as a novel marker of inflammatory response and metabolic disorders. PHR, a composite index combining platelet count and HDL-C levels, reflects both systemic inflammatory status and thrombotic tendency [14]. Recent studies have demonstrated strong associations between PHR and various chronic conditions, including hypertension, heart failure, and diabetes, highlighting its potential as a predictor of cardiometabolic diseases [15-17]. Obesity-induced chronic low-grade inflammation represents a key risk factor in the pathogenesis and progression of metabolic disorders [4]. A retrospective study [18] demonstrated that PHR can be utilized as a valid marker in obese patients to reliably predict the presence of type 2 diabetes. Alice Marra et al. [19] showed that five inflammatory indices derived from complete blood cell counts, including PHR, have good validity in predicting MetS in severely obese adults, providing clinicians with useful tools to assess obesity-related metabolic risk. While earlier research has examined the association between PHR and metabolic diseases such as hypertension and diabetes, there remains a paucity of information regarding the relationship between PHR and MetS and its defined overweight/obesity phenotypes. Consequently, the present study aims to analyze the relationship between PHR and MetS, as well as metabolic obesity/overweight phenotypes, with the goal of providing deeper insights for clinical practice.

Comment 2: The scientific terminology is inappropriately reformulated thorough the text leading to confusions. For example, instead of saying “Classification of metabolic overweight /obesity phenotypes” use simple terms like: classification according to metabolic status and BMI.

Response: Thank you for your valuable comments. We have made the following changes:

The individuals were categorized into four groups according to their BMI and metabolic status [24]:

(1)MHNW defined as BMI < 24 kg/m2 and unhealthy metabolic status;

(2)MHO defined as BMI ≥ 24 kg/m2and healthy metabolic status;

(3)MUNW defined as BMI < 24 kg/m2and unhealthy metabolic status;

(4)MUO defined as BMI ≥ 24 kg/m2and unhealthy metabolic status.

Comment 3: The research methodology is misleading because of the choice of potential confounding factors for adjustments. Some confounders used by the authors cannot be consider as such because they were already used to classify the participants as MU or MH (Adult Treatment Panel III (ATP III) criteria (Blood pressure, diastolic and systolic, Fasting plasma glucose; High-density lipoprotein cholesterol (HDL-C), and Triglycerides) or to classify the participant as Obese and non-obese (BMI). So, they cannot be used as confounders. Hence,

In sections 3.2 and 3.3, as well as in the corresponding Tables 2 and 3, only two models should be kept: the unadjusted model and a model adjusted for these factors: gender, age, BMI, FMR. ALT, AST, UA, tobacco use, alcohol use and NAFLD.

The same comment can be done for Table 4 regarding the adjustment.

In subgroup analyses, do not include comorbidity conditions (hypertension, diabetes and NAFLD) because they are already associated to MU status.

Response:Thank you for your in-depth review and valuable comments on our study. We greatly appreciate your feedback regarding the selection of potential confounders. Based on your suggestions, we have re-conducted the data analysis.

Comment 4: The comparison of PHR across the 4 phenotypes (Figure 1) should be done by ANOVA test because the Kruskal-Wallis test is a non-parametric test and the sample size is considerable.

Response:Thank you for your review of our paper and your valuable comments. In response to your suggestion that the ANOVA test should be used to compare PHR among the four phenotypes (MHNW, MHO, MUNW, MUO), we would like to provide further clarification regarding our choice of analysis.

We performed multiple normality tests (e.g., Anderson-Darling, Cramer-von Mises, Lilliefors, Pearson chi-square, and Shapiro-Francia tests), all of which yielded p-values of less than 0.000001. The null hypothesis was strongly rejected, indicating that the data did not conform to a normal distribution. This finding suggests that we should not assume a normal distribution for subsequent statistical analyses (See Fig.S1�.

ANOVA is a parametric test that assumes the data are normally distributed and that the variances among groups are equal. Since our data did not meet these assumptions, using ANOVA could have resulted in inaccurate and misleading results.

The Kruskal-Wallis test is a nonparametric test suitable for comparing the medians of three or more independent samples, particularly when the data do not follow a normal distribution. Given that our data deviated significantly from normality, the Kruskal-Wallis test was more effective in handling these data, ensuring the validity and reliability of the analysis results.

Based on the analysis above, we decided to use the Kruskal-Wallis test instead of ANOVA for comparing PHR among different phenotypes. This choice was made to ensure that our statistical analysis aligned with the actual distribution characteristics of the data, thereby increasing confidence in the results.

Of course, we also conducted an ANOVA analysis based on your suggestions, and the results are presented in Fig.S2. Both analytical methods demonstrated consistent differences between the groups.

We are very grateful for your attention and support regarding our study. If you have any further questions or suggestions, we would be happy to discuss them.

Comment 5: Table 1: please mention in the title “by PHR tertiles”

Response: Thank you for your valuable comments. It has been added to the article:

Table 1. Clinical characteristics of the study participants by PHR tertiles.

Comment 6: Put the section 3.3 (Smooth curve fitting and threshold effect analysis) after the section 3. 2

Response: Thank you for your valuable comments. The data were reanalyzed, and the manuscript was substantially revised.

Comment 7: Use the same abbreviations: sometimes we can read MHNW and sometimes MHNO.

Response: Thank you for your valuable feedback regarding the abbreviations. Based on BMI categories and metabolic status�we defined and distinguished four metabolic overweight/obesity phenotypes: metabolically healthy with normal weight (MHNW), metabolically unhealthy with normal weight (MUNW), metabolically healthy with overweight or obesity (MHO), and metabolically unhealthy wi

---

## [Editor Report · Decision Letter 1]

11 Mar 2025

Association of the platelet-to-high-density lipoprotein cholesterol ratio(PHR) with metabolic syndrome and metabolic overweight/obesity phenotypes: A study based on the Dryad database

PONE-D-25-00775R1

Dear Dr. Ping Zhang,

We’re pleased to inform you that your manuscript has been judged scientifically suitable for publication and will be formally accepted for publication once it meets all outstanding technical requirements.

Kind regards,

Marwan Al-Nimer

Academic Editor

PLOS ONE
---

## [Editor Report · Acceptance letter]

PONE-D-25-00775R1

PLOS ONE

Dear Dr. Zhang,

I'm pleased to inform you that your manuscript has been deemed suitable for publication in PLOS ONE. Congratulations! Your manuscript is now being handed over to our production team.

Kind regards,

on behalf of

Dr. Marwan Al-Nimer

Academic Editor

PLOS ONE